# Occupational Radiation Exposure and Thyroid Nodules in Healthcare Workers: A Review

**DOI:** 10.3390/ijms26136522

**Published:** 2025-07-07

**Authors:** Aikaterini Andreadi, Stella Andreadi, Marco Cerilli, Federica Todaro, Massimiliano Lazzaroni, Pietro Lodeserto, Marco Meloni, Cristiana Ferrari, Alfonso Bellia, Luca Coppeta, Andrea Magrini, Davide Lauro

**Affiliations:** 1Section of Endocrinology and Metabolic Diseases, Department of Systems Medicine, University of Rome Tor Vergata, 00133 Rome, Italy; federica.todaro@uniroma2.it (F.T.); massimiliano.lazzaroni95@gmail.com (M.L.); lodesertopietro17@gmail.com (P.L.); meloni.marco@libero.it (M.M.); bellia@med.uniroma2.it (A.B.); 2Endocrinology and Diabetology Clinic, Department of Medical Sciences, Foundation Policlinico Tor Vergata, 00133 Rome, Italy; marco.cerilli@ptvonline.it; 3Department of Biomedicine and Prevention, University of Rome Tor Vergata, 00133 Rome, Italy; stella.p.and@gmail.com (S.A.); cristiana.ferrari@ptvonline.it (C.F.); luca.coppeta@ptvonline.it (L.C.); andrea.magrini@uniroma2.it (A.M.); 4Unit of Medicine of Work, Foundation Policlinico Tor Vergata, 00133 Rome, Italy

**Keywords:** thyroid nodules, radiation exposure, healthcare workers, prevalence and radiation safety

## Abstract

Thyroid nodules are a common clinical finding, with their prevalence influenced by multiple environmental and occupational factors, including exposure to ionizing radiation. Healthcare workers, particularly those operating in radiology, nuclear medicine, interventional cardiology, and radiation oncology, are potentially at increased risk due to chronic low-dose radiation exposure. This review aims to evaluate the current evidence regarding the association between occupational radiation exposure and the development of thyroid nodules among healthcare professionals. The findings suggest a higher prevalence of thyroid nodules in radiation-exposed workers compared to the general population, although data heterogeneity and methodological limitations exist. Factors such as the duration of exposure, radiation protection practices, and frequency of monitoring play critical roles in modulating the individual risk. While some studies report no significant difference in malignancy rates, the increased detection of nodules underlines the need for regular thyroid surveillance in at-risk populations. Further longitudinal and multicentric studies are warranted to clarify the causality and guide preventive strategies. This review highlights the importance of occupational health protocols, including radiation shielding and periodic thyroid evaluation, in safeguarding the long-term endocrine health of healthcare workers.

## 1. Introduction

Thyroid nodules are a prevalent clinical finding, with their occurrence influenced by various environmental and occupational factors. Among these, exposure to ionizing radiation stands out as a significant risk factor, particularly in healthcare settings where professionals are routinely exposed to low-dose radiation. The thyroid gland’s sensitivity to radiation-induced damage underscores the importance of understanding the implications of such occupational exposures.

Healthcare workers, especially those in radiology, nuclear medicine, and interventional cardiology, are frequently subjected to chronic low-dose ionizing radiation. Studies have indicated a higher prevalence of thyroid nodules among these professionals compared to the general population. A recent cross-sectional study found a 23.3% nodule prevalence among radiology staff [1], underscoring the occupational risk [1,2].

Moreover, research has explored the functional impact of radiation exposure on the thyroid gland. Alterations in thyroid hormone levels, such as decreased free triiodothyronine (fT3) and free thyroxine (fT4), alongside increased thyroid-stimulating hormone (TSH), have been observed in exposed individuals, suggesting subclinical hypothyroidism [3]. These functional changes may precede or accompany the development of structural abnormalities like nodules.

The relationship between occupational radiation exposure and the thyroid cancer risk has also been a subject of investigation. A meta-analysis encompassing over three million individuals found that occupational radiation exposure was associated with a 67% increased risk of thyroid cancer, with male workers exhibiting a slightly higher risk than females [4]. However, the data remain heterogeneous, and further research is needed to establish a definitive causal link.

In Italy, legislative measures such as Legislative Decree 101/2020 have been implemented to safeguard workers from radiation exposure. This decree classifies exposed workers based on their annual radiation doses and requires regular health monitoring, including thyroid evaluations, to identify early signs of radiation-induced damage.

Despite these protective measures, the existing literature presents conflicting findings regarding the relationship between occupational radiation exposure and thyroid nodules. Some studies report a significant association, while others find no substantial differences between exposed and non-exposed groups. This inconsistency highlights the need to comprehensively review the current evidence to clarify the potential occupational risks and inform preventive strategies.

## 2. Thyroid Physiology and Radiation Sensitivity

The thyroid gland plays a crucial role in regulating metabolism, growth, and development by secreting triiodothyronine (T3) and thyroxine (T4) hormones. Its unique ability to concentrate iodine makes it particularly sensitive to ionizing radiation. Even low-dose exposure can disrupt hormone synthesis and cellular integrity. Radiation may induce oxidative stress, DNA damage, and mutations in thyroid tissue. These changes can lead to functional alterations, nodule formation, or malignancy. Understanding these mechanisms is essential for assessing occupational risks and developing preventive strategies.

### 2.1. Structure and Function of the Thyroid Gland

The thyroid gland is a vital endocrine organ located in the anterior neck, spanning the vertebral levels from C5 to T1. It consists of two lateral lobes connected by a central isthmus, which gives it a characteristic butterfly shape. Anatomically, the gland is positioned anterior to the trachea and inferior to the larynx, enveloped by the pretracheal fascia within the visceral compartment of the neck [5] (Figure 1).

Each thyroid lobe measures approximately 5 cm in length, 3 cm in width, and 2 cm in thickness, while the isthmus is about 1.25 cm in height and width. The gland is highly vascularized, receiving arterial blood from the superior and inferior thyroid arteries and draining through the superior, middle, and inferior thyroid veins. Lymphatic drainage is directed toward the prelaryngeal, pretracheal, and paratracheal lymph nodes [6].

Microscopically, the thyroid is made up of numerous spherical structures known as follicles. These follicles are lined by a single layer of cuboidal epithelial cells referred to as follicular cells, which encircle a central lumen filled with colloid. The colloid is rich in thyroglobulin, the precursor protein for synthesizing thyroid hormones. The primary hormones produced are T4 and T3, which are crucial to regulating the basal metabolic rate, oxygen consumption, thermogenesis, and overall energy metabolism [7].

Scattered among the follicular cells are parafollicular cells, also known as C cells, which secrete the hormone calcitonin. Calcitonin is involved in calcium homeostasis and primarily functions by inhibiting osteoclast activity, leading to reduced calcium levels in the blood [8].

The production and release of thyroid hormones are tightly regulated by the hypothalamic–pituitary–thyroid (HPT) axis. The hypothalamus secretes thyrotropin-releasing hormone (TRH), which stimulates the anterior pituitary gland to release thyroid-stimulating hormone (TSH). TSH, in turn, acts on the thyroid gland to promote the synthesis and secretion of T3 and T4. This endocrine loop is modulated by negative feedback, where elevated thyroid hormone levels inhibit further secretion of TRH and TSH to maintain homeostasis [9].

The thyroid gland’s structure and hormonal function are crucial for metabolic regulation, growth, and the calcium balance, highlighting its central role in endocrine physiology.

### 2.2. Mechanisms of Radiation-Induced Thyroid Damage

The thyroid gland is particularly susceptible to ionizing radiation, which can lead to a spectrum of functional and structural alterations. These changes result from various mechanisms, including DNA damage, oxidative stress, inflammation, and immune dysregulation.

#### 2.2.1. DNA Damage and Repair Mechanisms

Ionizing radiation induces DNA double-strand breaks (DSBs), a critical form of genetic damage. The repair of DSBs primarily involves two pathways: non-homologous end joining (NHEJ) and homologous recombination (HR). In thyroid cells, errors in these repair processes can lead to chromosomal rearrangements, such as RET/PTC gene fusions, which are implicated in radiation-induced papillary thyroid carcinoma (PTC) [10]. Studies have demonstrated that radiation exposure can cause specific gene rearrangements, contributing to thyroid carcinogenesis [11].

#### 2.2.2. Oxidative Stress and Cellular Damage

Radiation exposure generates reactive oxygen species (ROS), leading to oxidative stress that damages cellular components, including lipids, proteins, and nucleic acids. This oxidative damage can impair thyroid cell function and contribute to the development of hypothyroidism [12]. Moreover, oxidative stress can activate signaling pathways that promote inflammation and apoptosis, further exacerbating thyroid tissue injury [13].

#### 2.2.3. Inflammatory Responses and Autoimmunity

Radiation-induced damage can trigger inflammatory responses characterized by the release of cytokines such as interleukin-1 (IL-1), tumor necrosis factor-alpha (TNF-α), and interferon-gamma (IFN-γ). These cytokines can disrupt thyroid hormone synthesis and promote autoimmune reactions against thyroid antigens, potentially leading to autoimmune thyroiditis [5]. Evidence suggests that radiation exposure may increase the risk of developing autoimmune thyroid diseases, including Hashimoto’s thyroiditis and Graves’ disease [14].

#### 2.2.4. Functional Impairments and Hypothyroidism

Radiation can impair the thyroid’s ability to uptake iodine and synthesize hormones, leading to functional deficits. Hypothyroidism is a common consequence of radiation exposure, particularly in individuals receiving head and neck radiotherapy. The risk of hypothyroidism correlates with the radiation dose received by the thyroid gland [15]. Additionally, radiation-induced damage to the hypothalamic–pituitary–thyroid axis can disrupt regulatory mechanisms, further contributing to thyroid dysfunction [16].

#### 2.2.5. Age- and Dose-Dependent Effects

Children are more sensitive to radiation-induced thyroid damage due to higher mitotic activity and longer post-exposure life expectancy. Epidemiological studies, such as those following the Chernobyl nuclear accident, have shown a higher incidence of thyroid cancer and nodular disease in children exposed to radiation [17]. The risk of thyroid damage increases with higher radiation doses, and even low-dose exposures can have significant effects, especially in younger populations [15].

## 3. Occupational Radiation Exposure in Healthcare Settings

Occupational radiation exposure in healthcare settings arises from various sources, including diagnostic and interventional radiology, nuclear medicine, radiation oncology, dental practices, and veterinary practices. Implementing comprehensive radiation protection programs, including personal protective equipment, proper training, and adherence to safety protocols, is vital to minimize the exposure risks for healthcare workers.

### 3.1. Sources of Exposure

Healthcare professionals are routinely exposed to ionizing radiation through various medical procedures and equipment. Understanding the primary sources of this exposure is crucial to implementing effective radiation protection strategies (Figure 2).

Diagnostic radiology: Diagnostic radiology is a significant source of occupational radiation exposure. Procedures such as X-rays, computed tomography (CT) scans, and fluoroscopy emit ionizing radiation that can affect healthcare workers, especially radiologic technologists and radiologists. The increasing use of these imaging modalities has led to a rise in occupational exposure levels. A study highlighted that radiologic technologists are among the earliest occupational groups exposed to ionizing radiation and represent a large segment of the working population exposed to radiation from human-made sources [18].

Interventional radiology and cardiology: Interventional procedures, including angiography and cardiac catheterization, involve prolonged fluoroscopy, leading to higher radiation doses for medical staff. Operators and assisting personnel are exposed to scatter radiation, which can be significant over time. Research indicates that operator radiation exposure increases with certain procedural factors, emphasizing the need for protective measures [19].

Nuclear medicine: Nuclear medicine involves the use of unsealed radioactive sources for diagnostic and therapeutic purposes. Healthcare workers in this field are exposed to radiation during the preparation, administration, and disposal of radiopharmaceuticals. A study assessing occupational exposure in nuclear medicine practices found that medical personnel represent the largest group of workers occupationally exposed to ionizing radiation, although the health risks associated with low-dose exposure continue to be investigated [20].

Radiation oncology: In radiation oncology, staff are exposed to ionizing radiation during the operation of linear accelerators and brachytherapy procedures. While modern equipment includes safety features, accidental exposures can occur, particularly during equipment maintenance or source handling. A review of occupational radiation exposure in medical settings emphasizes the need for stringent safety protocols to minimize the risks [21].

Dental radiography: Dental professionals are exposed to radiation during intraoral and panoramic radiographic procedures. Although the doses are generally low, repeated exposure without proper shielding can accumulate over time. An article on occupational hazards in dentistry discusses the risks associated with radiation exposure and the importance of protective measures [22].

Veterinary radiology: Veterinary staff performing diagnostic imaging on animals are also at risk of radiation exposure. The need to manually restrain animals during procedures can lead to increased exposure to scattered radiation. Studies suggest that adherence to radiation safety protocols is essential in veterinary settings to protect staff.

### 3.2. Radiation Safety Regulations (e.g., Legislative Decree 101/2020)

Occupational exposure to ionizing radiation in healthcare settings necessitates stringent regulatory frameworks to safeguard workers’ health. In Italy, Legislative Decree 101/2020 implements the European Directive 2013/59/Euratom, establishing basic safety standards for protection against the dangers arising from exposure to ionizing radiation. This decree emphasizes the principles of justification, optimization, and dose limitation to minimize the radiation risks.

The decree mandates that occupational exposure should not exceed an effective dose of 20 millisieverts (mSv) per year, averaged over defined periods, and sets equivalent dose limits for specific organs, including 20 mSv per year for the lens of the eye and 500 mSv per year for the skin and extremities. These limits are designed to prevent deterministic effects and reduce the probability of stochastic effects, such as cancer and hereditary disorders.

Healthcare facilities are required to implement comprehensive radiation protection programs, including risk assessments, monitoring of exposure levels, maintenance of exposure records, and provision of appropriate training and protective equipment. Regular health surveillance is also mandated to detect any early signs of radiation-induced health effects, particularly in radiosensitive organs like the thyroid gland.

Studies have highlighted the importance of such regulations. For instance, a cross-sectional study conducted in Southern Italy found a higher prevalence of thyroid diseases among healthcare workers exposed to low-level ionizing radiation compared to non-exposed workers, underscoring the need for effective radiation protection measures [23].

Furthermore, research indicates that even low-dose occupational exposure can lead to alterations in thyroid function. A study assessing thyroid hormones and ultrasonographic abnormalities in medical staff occupationally exposed to ionizing radiation revealed significant differences in the thyroid parameters compared to the control groups. [24]

These findings reinforce the critical role of regulatory frameworks, such as Legislative Decree 101/2020, in establishing safety standards and promoting practices that protect healthcare workers from the potential adverse effects of ionizing radiation.

## 4. Epidemiological Evidence

### 4.1. Prevalence of Thyroid Nodules in Radiation-Exposed Workers

Several epidemiological studies have reported a higher prevalence of thyroid nodules in workers exposed to chronic low-dose radiation. A cross-sectional study of nuclear power plant workers in Korea, for instance, found that 30.6% of workers had thyroid nodules, a rate significantly higher than in the general population [25]. Similarly, research conducted among healthcare professionals working in interventional radiology showed an increased risk of thyroid nodules compared to matched controls [26]. The elevated prevalence is often attributed to the radiosensitive nature of thyroid tissue, mainly when exposure occurs over prolonged periods.

The pathophysiology underlying radiation-induced thyroid nodules involves DNA damage, cellular oxidative stress, and mutations in oncogenes such as RAS and RET/PTC rearrangements [11]. These genetic changes can lead to cellular proliferation and nodule formation, even in the absence of overt thyroid dysfunction. Although many nodules remain benign, the risk of malignancy, particularly papillary thyroid carcinoma, is also elevated in radiation-exposed groups [27].

The age at exposure, cumulative radiation dose, and duration of occupational exposure are key determinants influencing the prevalence of nodules. Younger individuals, particularly those exposed before the age of 20, are more susceptible to radiation-induced thyroid changes [28]. Moreover, repeated low-dose exposures, such as those encountered in medical-imaging professions, contribute to a cumulative effect that may promote the development of nodules over time.

Ultrasound surveillance has been instrumental in detecting subclinical thyroid nodules in radiation-exposed workers. High-resolution imaging enables early diagnosis, risk stratification, and monitoring of nodule growth. In occupational settings, the implementation of regular thyroid screening protocols has led to earlier detection and improved management of potential malignancies [29].

Preventive strategies include the use of personal protective equipment (PPE), adherence to radiation safety guidelines, and regular dose monitoring. Lead collars and thyroid shields, for instance, have been shown to significantly reduce thyroid exposure in interventional radiology settings. Additionally, education and training on radiation safety are crucial for minimizing unnecessary exposure and associated thyroid risks.

### 4.2. Comparative Studies with Non-Exposed Populations

Comparative studies between radiation-exposed workers and non-exposed populations have provided critical insights into the impact of occupational radiation exposure on thyroid health, particularly the development of thyroid nodules. These studies are essential for identifying the excess risk attributable to radiation, differentiating it from the background incidence, and informing occupational safety guidelines.

Numerous cross-sectional and cohort studies have shown a significantly higher prevalence of thyroid nodules in radiation-exposed workers compared to matched non-exposed controls. For example, a study conducted in Ukraine comparing Chernobyl clean-up workers with non-exposed individuals found a markedly increased rate of thyroid nodules and functional abnormalities in the exposed group, even after controlling for age, sex, and iodine intake [30]. Similarly, a Japanese cohort study that followed Fukushima emergency workers demonstrated a statistically significant increase in the thyroid nodule prevalence compared to population-based controls from non-contaminated regions [31].

In a comparative study of interventional radiologists and administrative hospital workers, the exposed group had more than double the rate of thyroid nodules, even after adjusting for confounding factors such as smoking, alcohol use, and family history of thyroid disease [32]. These findings underscore the cumulative effect of low-dose chronic exposure in occupational settings.

Notably, studies also report differences in the characteristics of the nodules found in exposed versus non-exposed individuals. Radiation-related nodules tend to be smaller but more numerous and may present with suspicious features on ultrasound, such as microcalcifications, irregular margins, and increased vascularity, which are associated with a higher malignancy risk [33]. While most thyroid nodules are benign, comparative studies have identified a higher proportion of malignant nodules in exposed populations, raising concerns about radiation-induced carcinogenesis [28].

Prospective studies have provided further evidence of the long-term effects of radiation exposure. In a 10-year follow-up of medical radiologic technologists in the United States, the incidence of thyroid nodules was significantly higher in workers with greater cumulative radiation doses compared to matched controls with minimal or no occupational exposures [28,34]. This dose–response relationship strongly supports a causal link between occupational radiation and thyroid pathology.

These comparative studies also emphasize the importance of early detection and surveillance. Non-exposed populations generally do not require routine thyroid screening unless clinically indicated. In contrast, radiation-exposed workers benefit from periodic thyroid ultrasonography to detect nodules at an early stage, enabling prompt intervention when necessary.

### 4.3. Dose–Response Relationships

Understanding the dose–response relationship between radiation exposure and the development of thyroid nodules is crucial to assessing the risk among exposed workers. The dose–response relationship refers to the correlation between the amount of radiation absorbed by the thyroid and the subsequent biological effects, particularly the occurrence of nodular changes or malignancy. Numerous studies have demonstrated that even low to moderate levels of radiation exposure, when sustained over time, can significantly increase the risk of thyroid nodules and cancer, with a generally linear or linear–quadratic pattern depending on the dose range and age at exposure [11].

The thyroid gland is particularly radiosensitive, especially during periods of rapid growth or cellular turnover. Evidence from atomic bomb survivors in Hiroshima and Nagasaki shows a clear linear increase in the thyroid cancer risk with an increasing radiation dose, particularly in individuals exposed at a young age [13]. This relationship persists across various populations, including those occupationally exposed to hazardous substances. A study of U.S. radiologic technologists revealed that those with cumulative thyroid doses exceeding 100 mGy had a significantly higher prevalence of thyroid nodules and carcinomas compared to those with lower or negligible exposure [35].

The dose–response pattern appears to be most pronounced for doses between 50 and 500 mGy. Below 50 mGy, the risk remains elevated, but establishing statistical significance can be challenging due to background noise and confounding factors [36]. Nevertheless, some models suggest that no clear threshold exists, and even very low doses might carry a risk, supporting the “linear no-threshold” (LNT) model of radiation carcinogenesis [37].

Interestingly, the nature of the exposure—whether acute or chronic—has an impact on the magnitude of the risk. Chronic, low-dose exposures, as seen in healthcare and nuclear industry workers, may result in persistent DNA damage and cellular alterations due to the limited time available for cellular repair mechanisms, thereby contributing to a cumulative risk [38]. This differs from acute exposures, such as those from nuclear accidents, where high doses delivered over a short period may result in more immediate and severe thyroid damage.

Age and sex also modulate the dose–response effect. Children and adolescents are particularly vulnerable, with studies showing a significantly higher risk of thyroid abnormalities per unit of radiation dose compared to adults [39]. Women are also generally at higher risk, potentially due to hormonal influences on thyroid tissue sensitivity [40].

Advancements in dosimetry have enabled more accurate estimates of individual radiation doses to the thyroid, thereby improving the precision of dose–response analyses. Modern occupational safety programs incorporate these data to establish dose limits, recommend thyroid shielding, and enforce periodic health surveillance for exposed workers [41]. These efforts are aligned with the guidance provided in ICRP Publication 146, which underscores the need for organ-specific dose assessment and highlights the thyroid as a critical organ requiring targeted protection measures in occupational settings [42].

In summary, a positive dose–response relationship exists between exposure to ionizing radiation and the development of thyroid nodules. The strength and shape of this relationship depend on several factors, including the total dose, dose rate, exposure duration, age, and individual susceptibility. These findings underscore the importance of minimizing occupational exposure and implementing regular monitoring protocols to facilitate early detection and prevention.

## 5. Radiation Protection Strategies in the Workplace

### 5.1. Shielding Devices and Safety Protocols

The use of shielding devices and the implementation of safety protocols are fundamental strategies for reducing occupational radiation exposure, particularly among workers in medical, nuclear, and industrial settings. Given the established link between ionizing radiation and the development of thyroid nodules and other radiation-induced pathologies, protective measures are essential to safeguard the health of exposed personnel.

Shielding devices are physical barriers that attenuate radiation before it reaches sensitive tissues such as the thyroid gland. One of the most effective and widely used protective tools is the lead thyroid collar, also known as a thyroid shield (Table 1). These collars, typically made of lead-equivalent material, can reduce thyroid radiation doses by up to 90% when properly fitted and used consistently [43]. They are essential in interventional radiology, cardiology, and fluoroscopic procedures, where scattered radiation is a common occurrence.

In addition to thyroid shields, lead aprons, ceiling-suspended screens, and mobile lead barriers are integral components of radiation protection. Their effectiveness depends on appropriate positioning and regular maintenance to avoid wear-related degradation of the shielding capacity [44]. For example, studies have shown that damaged or poorly positioned protective gear can result in significant exposure to unshielded body parts, including the thyroid [45].

Safety protocols complement physical shielding by promoting best practices in minimizing the radiation dose. These include the principles of time, distance, and shielding: minimizing the time spent near radiation sources, maximizing the distance from the source, and using appropriate shielding to reduce exposure. Routine training and certification programs in radiation safety are crucial to ensure that workers understand and apply these principles correctly [19].

Occupational dose monitoring is another critical component of safety protocols. Personal dosimeters, often worn at the collar level outside the lead apron, allow for accurate tracking of the cumulative radiation exposure. When used in conjunction with real-time dose monitoring systems, these devices can help alert workers to areas with high doses and prompt immediate corrective actions [19].

Administrative controls also play a significant role in radiation safety. These measures include limiting access to high-radiation areas, rotating personnel to reduce individual exposure, and maintaining detailed records of exposure. A safety culture, driven by leadership commitment and continuous education, is equally crucial to ensuring protocol compliance [46].

Modern technological advancements have further enhanced safety. The use of pulsed fluoroscopy, dose-reduction software, and automatic exposure control in imaging equipment has led to significant reductions in patient and occupational exposure [47]. Additionally, integrating radiation safety measures into procedural planning, such as selecting lower-dose imaging techniques when clinically appropriate, helps reduce the overall risk.

Despite the availability of protective equipment and protocols, adherence varies across institutions and roles. Studies have reported suboptimal compliance with thyroid shield usage among healthcare workers, often due to discomfort, lack of availability, or underestimation of the risk [48]. Addressing these barriers through ergonomic design improvements, policy enforcement, and regular audits is essential to ensure comprehensive protection.

### 5.2. Education and Compliance Among Workers

Education and compliance among radiation-exposed workers play a central role in minimizing occupational health risks, particularly the development of thyroid nodules and other radiation-induced conditions. Effective radiation protection is not solely dependent on technology and shielding devices—it also relies heavily on informed behavior and adherence to safety protocols. Consequently, comprehensive education and training programs are critical to fostering a safety culture and ensuring compliance with established guidelines.

Studies have consistently demonstrated that workers with formal education in radiation protection exhibit higher compliance with safety measures and lower levels of occupational exposure [49]. Training typically covers fundamental radiation physics, biological effects of ionizing radiation, dose limits, protective equipment, and emergency response procedures. Recurrent training is also recommended to maintain knowledge over time and adapt to evolving technologies and regulations [41].

However, gaps in education and inconsistent training practices still persist in many institutions, leading to variable compliance levels across occupational roles. For instance, a survey among interventional radiology personnel revealed that while most workers were aware of the basic protection principles, only a minority regularly used thyroid shields and dosimeters [50]. In some cases, workers underestimate the cumulative risk of low-dose radiation exposure, particularly to radiosensitive organs like the thyroid, leading to poor adherence to protective practices.

A strong safety culture, led by management and supported by regular feedback mechanisms, can significantly improve compliance. Institutions that prioritize safety, provide accessible protective gear, and enforce clear policies tend to see higher compliance rates among workers [51]. Additionally, peer influence and departmental norms play a role—when safety practices are visibly and consistently followed by colleagues and supervisors, workers are more likely to adhere to them.

Barriers to compliance often include the discomfort or impracticality of protective equipment, lack of time, and insufficient enforcement of protocols. For example, thyroid collars may be perceived as cumbersome or restrictive, especially in fast-paced clinical environments [52]. Addressing these issues through the provision of ergonomically designed protective gear and by involving workers in safety planning can lead to improved usage and acceptance.

Real-time monitoring and feedback systems have also proven effective in increasing compliance. Technologies such as electronic dosimeters with immediate dose alerts can raise awareness and prompt behavioral adjustments during procedures [53]. Moreover, integrating radiation safety education into professional development and certification programs reinforces its importance and accountability across all levels of staff.

Auditing and reporting systems further enhance compliance by identifying lapses, tracking trends in exposure, and providing data-driven recommendations for improvement. When combined with recognition and reward programs for safe practices, these initiatives can motivate workers to consistently adhere to safety guidelines [54].

### 5.3. Institutional Responsibilities

Institutions that employ individuals to work in radiation-prone environments have a fundamental responsibility to protect their staff from occupational hazards, including ionizing radiation. Institutional responsibilities encompass the development, implementation, and enforcement of comprehensive radiation safety programs, ensuring compliance with national and international guidelines, and fostering a culture of safety that prioritizes the well-being of workers.

At the core of these responsibilities is the establishment of a structured radiation protection program. This includes conducting thorough risk assessments, defining safe work practices, setting dose limits, and regularly reviewing safety procedures to incorporate new scientific findings and technological advancements [55]. Institutions must also ensure access to adequate shielding devices, such as lead aprons, thyroid collars, and mobile protective barriers, along with regular equipment maintenance and quality control checks [19].

Training and continuous education are also key obligations. Institutions must provide mandatory radiation safety training tailored to the specific duties of staff members, whether in medical, industrial, or research settings. This education should be reinforced with periodic refresher courses to maintain awareness and compliance over time [56]. A failure to properly educate workers may result in unsafe practices and an elevated risk of thyroid disorders or other radiation-induced diseases.

Another critical component of institutional responsibility is occupational dose monitoring. Employers are required to supply personal dosimeters and track cumulative radiation exposure for all workers in designated radiation areas. Data collected from dosimetry must be reviewed regularly, and workers whose exposure approaches regulatory limits should be counseled and, if necessary, reassigned to lower-risk duties [57]. Institutions must also ensure that these dose records are stored securely and are accessible to both regulatory bodies and individual workers [58].

Policy enforcement is essential. Institutions must actively monitor compliance with safety protocols and implement disciplinary actions in cases of repeated or willful non-compliance. Regular audits and inspections by internal radiation safety officers or external regulatory agencies help identify gaps in safety practices and provide opportunities for corrective actions [59]. This includes inspecting protective gear, verifying proper use of equipment, and evaluating adherence to procedural protocols.

Furthermore, institutions must establish clear emergency response plans for accidental exposures or radiation incidents. These plans should be regularly reviewed and practiced through drills to ensure staff preparedness. Institutions also bear the responsibility of reporting significant overexposures to the relevant health and safety authorities and providing affected workers with appropriate medical evaluations and follow-up care [60].

Finally, fostering a positive radiation safety culture is perhaps the most important and challenging institutional responsibility. This includes promoting transparency, encouraging staff to report unsafe conditions without fear of reprisal, and recognizing departments or individuals that demonstrate exemplary adherence to safety practices [61].

In conclusion, institutions play a pivotal role in safeguarding radiation-exposed workers. By implementing effective safety protocols, monitoring exposure, and fostering a culture of compliance, institutions can significantly reduce the risk of radiation-induced conditions, such as thyroid nodules, and support the long-term health of their employees.

## 6. Discussion

This review highlights the growing concern regarding the prevalence of thyroid nodules among healthcare workers exposed to occupational ionizing radiation. The data consistently suggest that individuals employed in radiology, nuclear medicine, interventional cardiology, and radiation oncology face increased risks of both structural and functional thyroid abnormalities due to chronic low-dose radiation exposure. While the association between radiation and thyroid malignancy has been well established in high-dose settings, evidence from occupational exposures supports a similar, albeit subtler, risk pattern for thyroid nodule development.

Epidemiological studies reveal a clear trend: the prevalence of thyroid nodules is significantly higher in radiation-exposed workers compared to non-exposed populations. This finding is consistent across multiple geographic regions and professional settings. Notably, these nodules often exhibit ultrasound characteristics associated with a higher risk of malignancy, underlining the clinical relevance of early detection and risk stratification in exposed individuals.

The dose–response relationship provides further support for a causal link between radiation exposure and thyroid pathology. Even at doses below 100 mGy, particularly when exposure is cumulative and chronic, the risk appears to increase, especially in younger individuals and women. This supports the linear no-threshold (LNT) model, which posits that no level of radiation exposure is entirely without risk.

Preventive strategies are well documented but inconsistently applied across institutions. Despite the availability of shielding devices and well-defined safety protocols, compliance among workers remains variable. Factors such as inadequate training, lack of equipment availability, and underestimation of risk contribute to insufficient adherence. Institutional responsibilities, including mandatory training, routine surveillance, and strong enforcement of safety regulations, are therefore essential to mitigating these risks.

Emerging experimental platforms such as spatial transcriptomics applied to 3D thyroid spheroid models and immune-on-a-chip systems offer promising avenues to investigate localized molecular responses and immune-mediated mechanisms triggered by chronic low-dose radiation, thereby complementing epidemiological findings with mechanistic insights.

A critical limitation of the current evidence is the heterogeneity of the study designs, exposure assessment methods, and population characteristics. This variability limits the ability to draw definitive causal inferences and underscores the need for standardized, multicenter, longitudinal studies to further explore these associations.

## 7. Conclusions

Occupational exposure to ionizing radiation among healthcare workers is associated with a higher prevalence of thyroid nodules, reflecting the thyroid gland’s inherent radiosensitivity. While the majority of these nodules are benign, their presence may signal underlying radiation-induced damage and carry potential implications for malignancy risk. The data support a dose-dependent relationship and reinforce the importance of regular surveillance, particularly in high-risk subgroups.

To safeguard healthcare workers, it is essential to implement and enforce comprehensive radiation protection strategies, including the use of shielding devices, education and training programs, exposure monitoring, and institutional oversight. Strengthening compliance and promoting a culture of safety can substantially reduce the incidence of radiation-induced thyroid disorders.

Future research should focus on large-scale, prospective studies to better define exposure thresholds and refine risk stratification. Prospective cohort study designs represent a valuable approach for improving long-term risk assessment by enabling the systematic tracking of radiation exposure and thyroid outcomes over time in well-defined worker populations. In parallel, emerging innovations in wearable dosimetry offer promising tools for real-time exposure monitoring, enhancing both individual risk awareness and institutional safety practices. These efforts will be instrumental in shaping evidence-based guidelines and ensuring the long-term endocrine health of professionals exposed to radiation.

## Figures and Tables

**Figure 1 ijms-26-06522-f001:**
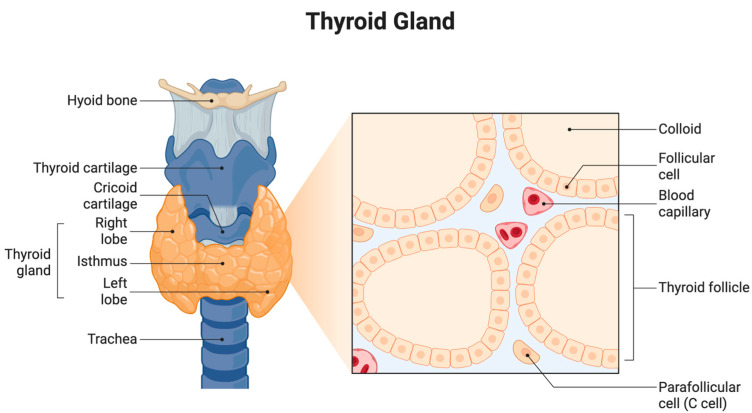
Anatomy and histology of the thyroid gland. The anatomical diagram shows the location of the thyroid gland, including the right and left lobes and the isthmus. The histological inset highlights thyroid follicles filled with colloid, surrounded by follicular cells, and includes parafollicular (C) cells and blood capillaries.

**Figure 2 ijms-26-06522-f002:**
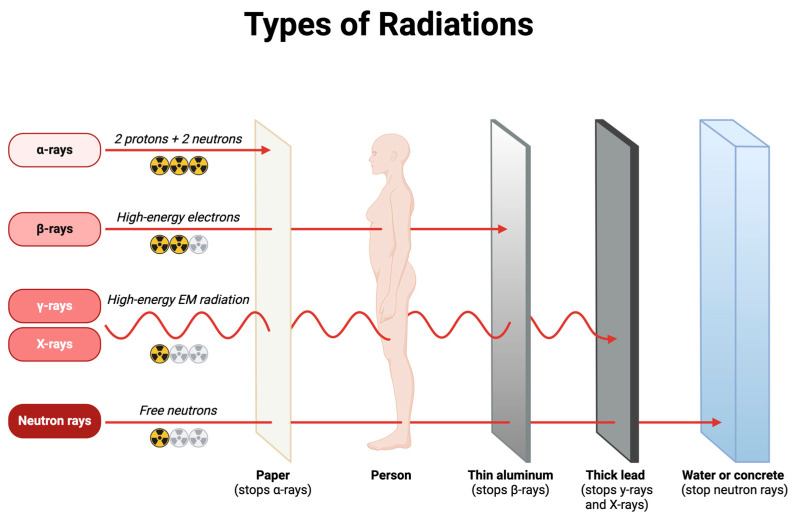
Penetration abilities of different types of radiation. Alpha, beta, gamma, X-rays, and neutron rays vary in their composition and ability to penetrate materials. Barriers such as paper, aluminum, lead, and concrete offer varying levels of protection depending on the type of radiation.

**Table 1 ijms-26-06522-t001:** Summary of key radiation protection strategies, their effectiveness in reducing thyroid exposure, and the typical levels of compliance in healthcare settings. The table highlights the gap between the available protective measures and real-world adherence.

Preventive Strategy	Description	Effectiveness	Reported Compliance
**Thyroid Shield (Collar)**	Lead-equivalent collar protecting thyroid from scatter radiation	Reduces dose by up to 90%	Often suboptimal (50–70%)
**Lead Apron**	Core protective garment for torso and organs	High when properly fitted	>90% in most clinical settings
**Dosimeter Use**	Worn at collar level to monitor cumulative radiation exposure	Enables dose tracking and alerts	Variable; often <70% consistently
**Real-Time Dosimetry**	Alerts workers to dose during procedures	Promotes active adjustment	Limited implementation
**Training Programs**	Mandatory radiation safety courses and refreshers	Increases awareness/compliance	Highly variable across institutions
**Administrative Measures**	Rotation, risk assessment, mandatory surveillance	Reduces exposure time	Generally enforced, but inconsistently
**Audits and Enforcement**	Institutional checks on protocol adherence and protective equipment use	Improves accountability	Rarely systematic outside research settings

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
