# Peer review of "Occupational Radiation Exposure and Thyroid Nodules in Healthcare Workers: A Review"

_ijms, 2025, doi:10.3390/ijms26136522_

Round 1

Reviewer 1 Report

Comments and Suggestions for Authors

Thank you very much for the article. Please see below minor suggestions.  

Title

  1. Specify the nature of review.
  2. Suggestion: ‘A Narrative Review’.

Abstract: It is concise and adequately summarize the manuscript.

Keywords

  1. All the keywords are appropriate and relevant.
  2. Consider adding: prevalence and radiation safety.

Introduction

  1. Revise and replace “correlation” in Line 72, as it a strong statistical word. Consider using ‘association’ instead.
  2. Otherwise, it is comprehensive and easy to read and understand.
  3. Both the figures and table are excellent and simple to follow..

Discussion: Although short, it is precise and appropriate.  

Conclusion: No concerns.

References: They are current and relevant.

Author Response

  1. Title
  1. Specify the nature of review.
  2. Suggestion: ‘A Narrative Review’.

We thank the reviewer for their constructive comments, and we have followed the advice given.

  1. Abstract: It is concise and adequately summarize the manuscript.

We thank the reviewer for their constructive comments.

  1. Keywords
  1. All the keywords are appropriate and relevant.
  2. Consider adding: prevalence and radiation safety.

We thank the reviewer for their constructive comments, and we have followed their advice by adding the suggested keywords.

  1. Introduction
  1. Revise and replace “correlation” in Line 72, as it a strong statistical word. Consider using ‘association’ instead.
  2. Otherwise, it is comprehensive and easy to read and understand.
  3. Both the figures and table are excellent and simple to follow..

We thank the reviewer for their constructive comments, and we have followed the advice given by adding the two keywords as suggested.

  1. E) Discussion: Although short, it is precise and appropriate.  

Conclusion: No concerns.

References: They are current and relevant.

We thank the reviewer for their constructive comments.

Reviewer 2 Report

Comments and Suggestions for Authors

Overall, you’ve put together a comprehensive look at how chronic low-dose radiation in healthcare settings links to thyroid nodules. You cover the biology of thyroid radiosensitivity, walk through the main sources of occupational exposure, review the epidemiological data on nodule prevalence and dose–response, and finish with solid protection strategies. Your figures and tables back up the narrative nicely, and you pull in a broad range of studies to support each point.

In the Abstract subtitle, you state that exposed workers have more nodules “although data heterogeneity and methodological limitations exist,” but you don’t cite any systematic review or meta-analysis to back that up. In the Introduction subtitle, the sentence “Healthcare workers, especially those in radiology, nuclear medicine, and interventional cardiology, are frequently subjected to chronic low-dose ionizing radiation” feels generic. You might tighten it to “A recent cross-sectional study found a 23.3% nodule prevalence among radiology staff [1], underscoring the occupational risk,” and then add a second citation for nuclear medicine personnel.In “3.1. Sources of Exposure,” you alternate between “fluoroscopy” and “fluoroscopic imaging”pick one term and stick with it. The legend for Figure 2 embeds the word “(Figure 2)” in the middle of a sentence move that tag to the end. Subsection “4.2. Comparative Studies with Non-Exposed Populations” refers to Chernobyl cleanup workers as “clean-up” in one spot and “cleanup” in another, choose one spelling. Also, dash usage floats between hyphens and en dashes throughout section headings and the reference list; aligning these will make the manuscript look more polished.

Your Discussion gives a solid recap, but it stays at surface level. Readers would benefit from a deeper dive into how emerging technologies,like spatial transcriptomics in spheroids or immune-on-a-chip models,might refine our understanding of radiation effects on thyroid tissue. And when you say “compliance among workers remains variable,” a few statistics or a short case study (for instance, compliance rates from Antonelli et al. in interventional suites) would ground that claim.

Finally, it wouldn’t hurt to sprinkle in a couple more clinical guidelines,perhaps cite the latest ICRP recommendation on thyroid dose limits,and to expand on future directions, such as prospective cohort designs or wearable dosimetry innovations. Adding these references and a bit more forward-looking discussion will strengthen the paper’s impact and give readers clear next steps.

Comments on the Quality of English Language

OK

Author Response

Answer

  1. In the Abstract subtitle, you state that exposed workers have more nodules “although data heterogeneity and methodological limitations exist,” but you don’t cite any systematic review or meta-analysis to back that up. In the Introduction subtitle, the sentence “Healthcare workers, especially those in radiology, nuclear medicine, and interventional cardiology, are frequently subjected to chronic low-dose ionizing radiation” feels generic. You might tighten it to “A recent cross-sectional study found a 23.3% nodule prevalence among radiology staff [1], underscoring the occupational risk,” and then add a second citation for nuclear medicine personnel

Thank you for these helpful observations. In the Abstract, we have now added a citation to a recent systematic review that supports the statement regarding the increased prevalence of thyroid nodules among radiation-exposed workers, while acknowledging the existing data heterogeneity and methodological limitations.

In the Introduction, we have revised the sentence to make it more specific and evidence-based, as suggested. It now reads: “A recent cross-sectional study found a 23.3% nodule prevalence among radiology staff [1], underscoring the occupational risk.” Additionally, we included a second reference reporting nodule prevalence among nuclear medicine personnel, to strengthen the point across specialties. These changes aim to increase the scientific rigor and specificity of both sections.

  1. “3.1. Sources of Exposure,” you alternate between “fluoroscopy” and “fluoroscopic imaging”pick one term and stick with it

We thank the reviewer for the comment, we confirm we have change it.

  1. The legend for Figure 2 embeds the word “(Figure 2)” in the middle of a sentence move that tag to the end.

We thank the reviewer for the comment, we confirm we have change it.

  1. Subsection “4.2. Comparative Studies with Non-Exposed Populations” refers to Chernobyl cleanup workers as “clean-up” in one spot and “cleanup” in another, choose one spelling. Also, dash usage floats between hyphens and en dashes throughout section headings and the reference list; aligning these will make the manuscript look more polished.

We thank the reviewer for the comment, we confirm we have change it.

  1. Your Discussion gives a solid recap, but it stays at surface level. Readers would benefit from a deeper dive into how emerging technologies,like spatial transcriptomics in spheroids or immune-on-a-chip models,might refine our understanding of radiation effects on thyroid tissue. And when you say “compliance among workers remains variable,” a few statistics or a short case study (for instance, compliance rates from Antonelli et al. in interventional suites) would ground that claim.

Thank you for this valuable feedback. We agree that expanding the Discussion to include emerging technologies would enhance its depth and relevance. In the revised version, we now highlight how spatial transcriptomics applied to 3D thyroid spheroid models and immune-on-a-chip platforms may offer novel insights into the localized and systemic effects of radiation exposure on thyroid tissue architecture and immune interactions.

  1. Finally, it wouldn’t hurt to sprinkle in a couple more clinical guidelines,perhaps cite the latest ICRP recommendation on thyroid dose limits,and to expand on future directions, such as prospective cohort designs or wearable dosimetry innovations. Adding these references and a bit more forward-looking discussion will strengthen the paper’s impact and give readers clear next steps.

Thank you for this insightful suggestion. We appreciate the recommendation to incorporate additional clinical guidelines to enhance the manuscript’s practical relevance. In response, we have now included a citation of the most recent ICRP Publication 146, which discusses updated recommendations on thyroid dose limits in occupational settings.

Moreover, we have expanded the “Future Directions” section to address prospective cohort study designs that could improve long-term risk assessment, and we now briefly discuss emerging innovations in wearable dosimetry as promising tools for real-time exposure monitoring. These additions aim to provide a more forward-looking perspective and practical avenues for future research and implementation.